# Influence of Respiratory Tract Infections on Vocabulary Growth in Relation to Child’s Sex: The STEPS Study

**DOI:** 10.3390/ijerph192315560

**Published:** 2022-11-23

**Authors:** Annette Nylund, Laura Toivonen, Pirjo Korpilahti, Anne Kaljonen, Viveka Lyberg Åhlander, Ville Peltola, Pirkko Rautakoski

**Affiliations:** 1Department of Speech and Language Pathology, Abo Akademi University, 20500 Turku, Finland; 2Department of Paediatrics and Adolescent Medicine, Turku University Hospital and University of Turku, 20521 Turku, Finland; 3Department of Psychology and Speech-Language Pathology, University of Turku, 20500 Turku, Finland; 4Statistics of the STEPS Study (Steps to the Healthy Development and Well-Being of Children), University of Turku, 20500 Turku, Finland

**Keywords:** acute otitis media, child’s sex, language development, MacArthur–Bates Communicative Development Inventory, respiratory tract infections, vocabulary

## Abstract

Common health issues have been less examined in studies of early language development, particularly in relation to the child’s sex. Respiratory tract infections, often complicated by acute otitis media, are common in children during the first years of life, when early vocabulary development takes place. The present study, conducted in Finland, aimed to investigate whether possible associations between recurrent respiratory tract infections, background factors, and vocabulary growth differ in boys and girls aged 13 to 24 months. The participants (*N* = 462, 248 boys and 214 girls) were followed for respiratory tract infections and acute otitis media from 0 to 23 months of age. The parents completed daily symptom diaries of respiratory symptoms, physician visits, and diagnoses. The expressive vocabulary was measured with parental reports. We found that recurrent respiratory tract infections were not associated with slower vocabulary development in boys or girls. In fact, boys with recurrent respiratory tract infections had more vocabulary growth during the second year than boys who were less sick. We found that vocabulary growth was associated differently with respiratory tract infections and background factors as a function of the child’s sex. The vocabulary growth of boys seems to be more influenced by environmental factors than that of girls.

## 1. Introduction

Early language development is influenced by the biological and environmental factors in a child’s life, such as the child’s own experiences and the social context in which the child grows up [1]. Emerging language is usually noticed by the words the child understands and produces, which is why vocabulary size has been a commonly used measure for assessing early language development [2,3,4]. Vocabulary and vocabulary growth play a major role in early language development between 1 and 2 years of age. It is only during the latter part of the second year that grammatical development starts to show in the child’s speech, and it becomes predominant after the second birthday [2]. This makes 13 to 24 months of age an interesting period during which to study early expressive vocabulary development. Furthermore, we know that language development during the first two years sets the trajectory for future language development [2,3]. There is still a lack of clarity related to the factors influencing early language development.

During the early years, a child is exposed to a variety of health challenges. One of these is respiratory tract infections (RTIs), which pose a common health threat to young children under 5 years of age [5] and are particularly salient during the first two years of life [6]. The symptoms of RTIs include rhinorrhea, cough, fever, wheezing, apathy, and eating problems. A common complication of viral respiratory infections in children is acute otitis media (AOM) [7], which is documented in 13–50% of children with RTIs [8,9]. As RTIs in the early years are frequent, they put a lot of pressure on the family, requiring repeated health care visits and the frequent use of antibiotics [6,10,11]. Respiratory infections are also easily transferred between the child, caregiver, and siblings [12]; thus, there may be sick parents taking care of sick children. A possible consequence for the family during periods of illness in the child is parental absence from work, with a remarkable loss of workdays [11]. Having sick infants and toddlers can be worrisome and stressful for parents. In addition to siblings, day care attendance has also been considered a risk factor for RTIs [8,9,13].

Possible relationships between RTIs alone, not AOM, and early language development have, to our knowledge, been addressed in only one study by Nylund et al. [14], which focused on the early receptive and expressive vocabulary size in relation to recurrent RTIs. The results indicated that children suffering from recurrent RTIs did not, at the group level, have fewer words at 13 or 24 months of age than children who were less sick. However, the study focused on children as a group, not boys vs. girls, thus any differences between the sexes would not have emerged. Earlier studies on the effects of RTIs on language development have mainly focused on the relationship between otitis media (OM, including AOM and otitis media with effusion) and young children’s language and vocabulary development but had conflicting results [15,16,17,18]. Some studies found few or no differences in early language development that could be attributed to OM or hearing loss associated with OM, while others found some correlations with AOM or OM [15,16,17,18]. Other studies focused on the consequences of hearing loss or possible hearing loss due to middle-ear infection, while others focused only on the associations between AOM or OM and language development. One explanation for the inconsistent results could be the methodological differences mentioned above or differences in the sample sizes in the studies or the selection of participants.

In addition, studies of early language development have reported differences between boys and girls, with a larger vocabulary size and a faster accelerating vocabulary during the early years in girls compared to boys [19,20,21]. All studies mentioned above studied the effects of RTIs or AOM/OM on language development in children as a group, not separately for boys and girls. This may conceal differences in how vocabulary growth in boys vs. girls is associated with RTIs or AOM. There are also differences in the incidence of RTIs and AOM in boys vs. girls. RTIs are reported during the first two years of life with a slightly higher predisposition [13,22] and more frequent hospital visits in boys than in girls [23]. Being a boy is also considered a risk factor for early AOM or OM [15,24,25]. Studies examining other possible factors that influence early language often also focus on a combined language factor for boys and girls. This may conceal possible differences in how language development relates to environmental factors in boys vs. girls. Nonetheless, a small number of studies have demonstrated differences in the influence of environmental factors on early language development as a function of the child’s sex. Barbu et al. [26] found differences in how a low socioeconomic (SES) background affected language development in boys and girls at ages 2.6 to 6.4. Moreover, Lankinen et al. [27] reported a stronger relationship between high paternal educational and occupational levels and expressive vocabulary size at 2 years of age in boys compared to girls. Environmental factors have also been associated differently with the early development of lexical categories in boys and girls [28].

When investigating early vocabulary growth, there are some background factors that need to be considered. The importance of the child’s family background and early experiences for later language development is well-established, (e.g., [29]). Some of the environmental factors that have been associated with early language development are, e.g., day care outside the home and particularly the parents’ SES [30,31]. Positive short-term effects of day care have been found in some studies [32]. However, several studies show positive effects in early language development for younger children staying at home [33,34,35]. Accordingly, there is still ambiguity in how day care influences early language development. Day care during the first years of life is a risk factor for RTIs and AOM, which show higher incidences in day care than among children cared for at home [13,36,37,38,39,40]. This means that children attending day care may have more RTI sick days, with possible effects on early language development. The inclusion of day care attendance as a background factor when analyzing early vocabulary development is therefore essential. Variations in vocabulary and language skills due to differences in SES background can be detected before two years of age [20,41]. However, many of the studies concerning SES and language or vocabulary development in young children focused on the education or occupation of the mother, (e.g., [20,31]), and there are also studies suggesting opposite results [28]. Previous studies have often used combined SES variables when analyzing early language development, sometimes including both maternal and paternal education and occupation in a combined factor and sometimes only including one or two of these variables, (i.e., [30,41,42]). However, as this may not give accurate results and this makes it difficult to compare between studies, Braveman et al. [43] and Duncan et al. [44] have proposed analyzing these factors separately. In the present study, we have chosen to focus on maternal and paternal education and occupational status. In the present study, we therefore separately studied the influence of the paternal and maternal levels of education and occupational status on vocabulary growth.

The overall aim of this longitudinal study was to investigate early expressive vocabulary growth in children between 13 and 24 months of age in relation to RTIs and AOM episodes. We particularly wanted to address the knowledge gap on possible associations between recurrent RTIs and AOM episodes with early vocabulary growth in boys vs. girls. In addition, day care attendance and parental educational and occupational levels were analyzed as background factors.

The aims were:

(a) to investigate whether vocabulary growth between 13 and 24 months of age differs in boys and girls with recurrent RTIs or AOM episodes compared to boys and girls without recurrent RTIs and AOM episodes.

We hypothesized that a consequence of recurrent RTIs and AOM episodes on vocabulary growth, possibly concealed when studying a combined measure for boys and girls, could be revealed if boys and girls were studied separately, as previous studies showed that language development can relate differently to environmental factors in boys vs. girls [26,27].

(b) to investigate vocabulary growth between 13 and 24 months of age in boys and girls in relation to background factors that are often associated with early language development (i.e., high maternal and paternal levels of education and occupation and whether or not the child had been in day care).

We hypothesized, based on previous research [26,27], that possible effects of background variables on vocabulary growth would differ in boys and girls.

## 2. Materials and Methods

The present study is a substudy of a prospective ongoing longitudinal observational cohort study, steps to the healthy development and well-being of children (the STEPS Study), with an eligible cohort of 9811 mothers and their 9936 children [45]. The Finnish Ministry of Social Affairs and Health and the Ethics Committee of the Hospital District of Southwest Finland approved the STEPS study in 2007. The families were recruited during pregnancy at maternity clinics or the delivery ward between January 2008 and April 2010 in the Hospital District of Southwest Finland. Altogether, 1797 Swedish- and Finnish-speaking families with their 1805 children from the eligible cohort chose to participate in the follow-up group of the study. The parents of the participating children gave written informed consent to the study.

### 2.1. Participants

Of 1805 liveborn participating children, a subgroup of 923 children (912 families) participated in the present substudy for the intensive follow-up of RTIs. The children were closely followed for RTIs, including AOMs, during the first two years of life [11]. In the present study, we aimed to longitudinally investigate expressive vocabulary development and included only children with available vocabulary data for both 13 and 24 months of age. In total, 184 children had language data for only one of the measuring points and thus were not included. The exclusion criteria were preterm delivery, missing or ambiguous data on gestational length, diagnoses or conditions connected with language problems, missing language data, and families who did not speak Finnish at home. After these exclusions, the sample size for analysis was 462 children (54% boys) from 460 families. The participants included two twin pairs (three girls and one boy). The recruitment procedure is described in Figure 1.

### 2.2. Data Collection

#### 2.2.1. Language Measures

Expressive vocabulary growth between 13 and 24 months of age was our primary interest. Expressive vocabulary was assessed with the Finnish version of the MacArthur–Bates Communicative Development Inventory (CDI) for infants (CDI-I) aged 8–16 months and for toddlers (CDI-T) aged 16–30 months [2,46,47] at two time points, 13 and 24 months of age.

The parents completed CDI questionnaires that had been sent to them either electronically or by post. The CDI questionnaires were completed by the mother, the father, or both. The vocabulary scores at 13 and 24 months of age were analyzed, but the main emphasis in the present study was on the difference in vocabulary size between 13 and 24 months, i.e., the growth in vocabulary. In the present study, the wording “vocabulary size” has been used when presenting the results from the CDI-I and CDI-T, although these scores do not cover the child’s whole vocabulary. The focus was on the expressive vocabulary part of the CDI, since this is the only part that occurs in both inventories. The expressive vocabulary part in the Finnish CDI-I consists of 380 words in 19 categories, and the expressive vocabulary part in the CDI-T consists of 595 words in 20 categories [46]. The parents of two of the study children did not mark anything in the expressive word section of the CDI-I, only marking the gestures section, which we took to mean that the children did not have any active word use at that point.

#### 2.2.2. Respiratory Tract Infections

The children were followed up for RTIs during the first two years. The parents documented all respiratory symptoms, associated physician visits, diagnoses, treatments, and the child’s absences from day care as well as any parental absence from work in a daily symptom diary [11]. Parents were advised to visit the STEPS study clinic if they suspected their child had an RTI, but they could also choose to visit another clinic. The findings of the physical examination at the study clinic were recorded on a structured form. Tympanometry (MicroTymp2, Welch Allyn, Skaneateles Falls, NY, USA) and pneumatic otoscopy (Macroview, Welch Allyn, Skaneateles Falls, NY, USA) were routinely used at the study clinic in diagnosing AOM. The diagnosis of AOM was determined according to the symptoms of acute RTI, the notation of effusion in the middle ear, and the signs of inflammation of the tympanic membrane. Respiratory tract infection was defined as the presence of rhinitis or cough (with or without fever or wheezing) documented in the symptom diary by the parents or as any acute RTI diagnosed by a physician. Episodes of RTI were counted as separate episodes if there was at least 1 day in between without any recording of symptoms. If respiratory symptoms persisted, recurrent AOMs were counted as separate episodes if there were at least 14 days between AOM diagnoses [11]. Recurrent RTIs and AOM episodes at 0–12 months and 13–23 months of age were considered risk factors in the present study.

#### 2.2.3. Demographic Data

During pregnancy, the parents completed questionnaires about systematic demographic data with a focus on family structure, family net monthly income, paternal and maternal education and occupation, and the family’s own health history. The questionnaires were answered by mothers at the 10th–15th weeks of gestation, by both parents separately at the 20th and 30th weeks of gestation, and by one of the parents or both when the child was 13 and 24 months of age. The mothers recruited at the delivery ward completed the questionnaire at that time. Child and parental background data previously linked to early language development were included in the study. Child variables included whether the child was in day care outside the home at 13 and 24 months of age. Parental variables were high paternal and maternal educational and occupational levels. The demographic questions in the questionnaires were in a multiple-choice format. The child variables were dichotomous. The parental variables that were not dichotomous in the questionnaire were made dichotomous to distinguish between high and low levels of education and occupation. Bachelor’s, master’s, and doctoral degrees were considered a high educational level, and a professional occupation (including managerial and specialist levels) was considered a high occupational level. The same criteria for high vs. low levels of education and occupational status were used in the study by Lankinen et al. [27].

### 2.3. Statistical Analysis

The primary outcome was vocabulary growth between the ages of 13 and 24 months; that of children with recurrent RTIs or AOM was compared to that of children without either. The primary exposures were the number of days with RTI symptoms at 0–12 months and 13–23 months of age. As AOM episodes are a common complication of RTIs, the number of AOM episodes was also studied at 0–12 months and 13–23 months of age. The number of days with RTI symptoms was adjusted with the duration of active follow-up using the symptom diary of each child. To identify children with recurrent RTIs, the upper 10th percentile of the number of days with RTI symptoms was used as a cut-off. Similarly, the upper 10th percentile of the number of AOM episodes was used as a cut-off for recurrent AOM. Day care attendance and high parental educational and occupational levels were included as background variables.

A chi-square analysis and a two-tailed *t*-test for independent samples were conducted to control possible systematic differences between the sample and the excluded participants as well as between children with and without recurrent RTIs. A two-tailed *t*-test for independent samples was used to compare the mean vocabulary growth between 13 and 24 months of age in children with and without recurrent RTIs or AOM and in relation to background factors. A two-way ANOVA with an interaction term was conducted with all background variables. Because a significant interaction was found for the interaction term “gender*maternal occupational status” (*p* < 0.001), it enabled us to analyze vocabulary growth separately for boys and girls. A linear regression was conducted to examine the predictive value of risk and background factors for vocabulary development. As the vocabulary growth rate can depend on the earlier vocabulary size, the vocabulary size at 13 months was added as a control variable in the linear regression. Another variable added in the regression as a control variable was the presence of siblings, as children with recurrent RTIs more often have older siblings [12]. An examination with a Spearman’s correlation analysis revealed that none of the independent variables were highly correlated with each other (*r* < 0.8). The collinearity statistics between the independent variables were all within the accepted tolerance limits of >0.2 and VIF < 10, and therefore the assumption of multicollinearity was met.

The data were analyzed using IBM SPSS Statistics Version 26.0 (Armonk, NY, USA) and SAS for Windows Release 9.4 (Cary, NC, USA). *p*-values of less than 0.05 were considered statistically significant.

## 3. Results

Of the total sample of children (*N* = 462), 61% were firstborn children, 26% had one sibling at birth, 9% had two siblings, and 4% had three to six siblings. The mean age of the mothers at delivery was 31.1 years, and the mean age of the fathers at delivery was 33.1 years. The mean net monthly income of the families at the birth of the child was slightly above the average net monthly income for households in Finland during the same period [48]. See Table 1 for the descriptive characteristics of the study population.

To examine whether there were any systematic differences between the included and excluded participants, we compared the background variables of the included families with those of nonparticipating families. There were no differences except in two cases. There was a difference in the age of the included and nonparticipating mothers (*mean* 31.1 vs. 30.6 years, *t*(853) = 2.210, *p* = 0.027). Other differences were that 61% of the included children were firstborn children compared to 50% of those not included (ꭓ^2^(1, *N* = 1805) = 16.032, *p* < 0.001) and 66% of the participating mothers were highly educated compared to 58% of those not participating (ꭓ^2^(1, *N* = 1726) = 10.482, *p* = 0.001).

There was wide variation in the early vocabulary size and expressive vocabulary development in the children, as reported by the parents. The mean expressive vocabulary size in all the study children was 8 words (*SD* 18) at 13 months and 295 words (*SD* 167) at 24 months. For vocabulary growth from 13 to 24 months of age, the mean was 287 (*SD* 163) words. A two-tailed independent *t*-test showed that the vocabulary size in the girls at 13 and 24 months of age was significantly larger than in the boys (*mean* 11 vs. 6 words, *p* = 0.005, 95% CI [−0.46, −0.10], Cohen’s *d* = −0.28 and *mean* 343 vs. 254 words, *p* < 0.001, 95% CI [−0.73, −0.36], Cohen’s *d* = −0.55, respectively). They also had a significantly larger vocabulary growth between 13 and 24 months (*mean* = 332, *SD* 150) compared to boys (*mean* = 249, *SD* 163; *p* < 0.001, 95% CI [−0.71, −0.34], Cohen’s *d* = −0.53) of the same age. See Figure 2a for the vocabulary growth in boys and girls and Figure 2b for the vocabulary growth in boys and girls with and without recurrent RTIs.

### 3.1. Respiratory Tract Infections in Study Sample

The mean numbers of days with RTI symptoms for boys and girls at 0–12 months of age were 48 days (*SD* 35) and 45 days (*SD* 37), respectively. The mean number of AOM episodes during the same period was one (*SD* 1) for both boys and girls. At 13–23 months of age, the mean numbers of days with symptoms of RTI for boys and girls were 60 (*SD* 51) and 55 (*SD* 44), respectively. The mean number of AOM episodes at 13–23 months of age was one (*SD* 2) for boys and girls. Children with ≥91 (age 0–12 months) and ≥120 (age 13–23 months) days with RTI symptoms (upper 10th percentile) were defined as having recurrent RTIs. Likewise, children with ≥3 (age 0–12 months) and ≥4 (age 13–23 months) AOM episodes were defined as having recurrent AOM. There were no significant differences in the incidences of recurrent RTIs and AOM episodes in boys and girls. In the study, recurrent RTIs were found during the first year in 27 boys and 19 girls, during the second year in 25 boys and 20 girls, and during the first two years in 29 boys and 17 girls. Similarly, recurrent AOM episodes were found during the first year in 23 boys and 15 girls and during the second year in 30 boys and 18 girls. In the current study, 19 children (11 boys and 8 girls) had both recurrent RTIs and recurrent AOM during their first two years of life. Children with siblings had significantly more RTIs (*p* < 0.001) and AOM episodes (*p* = 0.024) at age 0–12 months than those without siblings.

There was a significant correlation in boys between having recurrent RTIs during the first year and having recurrent RTIs during the second year (*r* = 0.179, *p* = 0.006). In both boys and girls, recurrent RTIs during the first year also correlated with recurrent AOM episodes during the second year (*r* = 0.310, *p* < 0.001 and *r* = 0.359, *p* < 0.001, respectively). An interaction was also found in boys and girls between having recurrent RTIs and recurrent AOM episodes during the second year (*r* = 0.282, *p* < 0.001 and *r* = 0.305, *p* < 0.001, respectively). In girls, recurrent AOM episodes during the first year of life correlated with recurrent RTIs during the second year (*r* = 0.229, *p* = 0.001).

Children with and without recurrent RTIs at ages 0–12 months and 13–23 months, respectively, did not significantly differ from each other concerning background factors except for being firstborn children and the level of parental education. Boys and girls with recurrent RTIs at age 0–12 months, were less likely to be firstborn children (18% and 10%, respectively) than children without (68% and 66%, respectively, *p* < 0.001). Boys with recurrent RTIs at age 0–12 months had a mother with a higher educational level (85%) more often than boys who were less sick (64%, *p* = 0.038). Girls with recurrent RTIs at age 13–23 months had fathers with a lower educational level (78%) more often than girls without recurrent RTIs (46%, *p* = 0.011). Girls with recurrent RTIs during the first year were less likely to be at day care outside the home during the second year (77.8%) compared to girls who were less sick (43.4%, *p* = 0.005). Girls with recurrent AOM episodes during the second year were more often in day care outside the home (83.3%) compared to girls who were less sick (52.1%, *p* = 0.009). The corresponding differences in relation to day care were not significant in boys.

### 3.2. Vocabulary Growth in Relation to Recurrent RTIs and Background Factors

Children with recurrent RTIs at age 0–12 or 13–23 months did not have a smaller vocabulary growth between 13 and 24 months of age than children who were less sick (Figure 2b). On the contrary, parents often reported more words for children with recurrent RTIs, but the difference was only significant in boys with recurrent RTIs at 13–23 months of age (*p* = 0.024, CI [−0.90, −0.06], Cohen’s *d* = −0.48). These boys had a significantly larger vocabulary growth than boys without recurrent RTIs. There was no significant difference in vocabulary growth if the boys had recurrent RTIs at 0–12 months of age. Girls with recurrent RTIs at 0–12 or 13–23 months of age did not differ significantly for vocabulary growth compared to girls who were less sick. There were no significant differences in vocabulary growth in children with recurrent AOM episodes compared to children without recurrent AOM episodes. This concerned vocabulary growth in both boys and girls with recurrent AOM episodes at 0–12 and 13–23 months of age. See Table 2.

When vocabulary growth was compared in relation to the included background factors, some differences emerged. However, this applied only to vocabulary growth in boys. Boys of highly educated fathers had a larger mean vocabulary growth (281, *SD* = 168 words) between 13 and 24 months of age than boys whose fathers had a lower level of education (225 words, *SD* = 154; *p* = 0.008, CI [0.09, 0.060], Cohen’s *d* = 0.34). Boys of professional (high occupational level) parents had greater vocabulary growth than those of nonprofessional parents. The mean vocabulary growth values of boys with professional vs. nonprofessional mothers were 278 (*SD* = 168) and 211 (*SD* = 152; *p* = 0.004, CI [0.13, 0.70], Cohen’s *d* = 0.42) words, respectively. The corresponding growth values for boys with professional vs. nonprofessional fathers were 262 (*SD* = 167) and 210 (*SD* =151; *p* = 0.025, CI [0.04, 0.60], Cohen’s *d* = 0.32) words, respectively. Corresponding significant differences associated with the educational or occupational levels of the parents were not found in the vocabulary growth of girls.

To examine possible predictors of vocabulary growth, linear regressions were conducted with risk (recurrent RTIs and AOM episodes) and background factors (high educational and occupational levels and day care attendance) separately for boys and girls. The vocabulary size at 13 months of age and having siblings were included as control variables. The linear regression model for boys was found to explain 12% of the variance in vocabulary growth between 13 and 24 months of age in boys (*F*(12, 138) = 2.70; *p* = 0.003, *R*^2^ = 0.19; *R*^2^*_Adjusted_* = 0.12). Recurrent RTIs at 13–23 months of age (*t* = 2.51, *p* = 0.013, *β* = 0.21) and vocabulary size at 13 months of age (*t* = 3.26, *p* = 0.001, *β* = 0.26) were significant predictors of vocabulary growth between 13 and 24 months of age in boys. The linear regression model for girls was not significant (Table 3).

## 4. Discussion

The purpose of the present study was to investigate whether expressive vocabulary growth between 13 and 24 months of age differs in children with recurrent RTIs compared to children who are less sick and whether recurrent RTIs relate differently due to the child’s sex. We also considered possible differences in early vocabulary growth in boys and girls in relation to high parental educational and occupational levels and day care attendance. We analyzed the influence of RTIs and AOM on vocabulary growth separately in boys (*n* = 248) and girls (*n* = 214). Our main results were that boys and girls with recurrent RTIs did not have smaller vocabulary growth than children without recurrent RTIs. However, boys with more days of RTIs at 13–23 months of age had significantly larger vocabulary growth than other boys. This phenomenon did not apply to girls. In addition, we found that vocabulary growth in boys related differently to recurrent RTIs, high paternal education, and high parental occupational levels compared to that in girls.

### 4.1. Vocabulary Growth

In line with earlier studies, the girls in the current study had a larger vocabulary size at 13 and 24 months of age, according to the CDI, and larger vocabulary growth between 13 and 24 months of age than the boys [19,20,21]. The vocabulary size at 13 months of age was a predictor of vocabulary growth in boys between 13 and 24 months of age. One interpretation could be that the trajectory for vocabulary growth is already settled at 13 months of age, meaning that boys with a small vocabulary size at this age will continue to have a more limited vocabulary size later on compared to boys who already have a larger vocabulary size at 13 months of age. However, it could also be the case that the boys with larger vocabulary growth between 13 and 24 months of age were in the so-called word spurt [49,50], explaining the larger vocabulary growth. However, as not all children develop their vocabulary following the pattern of a period of quick vocabulary expansion, this cannot be concluded [49].

### 4.2. Recurrent RTIs and Vocabulary Growth

Children with recurrent RTIs did not have smaller vocabulary growth compared to children who were less sick; nor did children with recurrent AOM episodes. This contradicts our hypothesis that a high rate of RTIs would be negatively associated with language development. As the second year is a very significant time in language development, the burden of recurrent RTIs does not seem to interfere with that development. Instead, recurrent RTIs at 13 to 23 months of age predicted better vocabulary growth in boys. One reason that recurrent RTIs did not limit vocabulary growth could be the opportunity for the study families to immediately access a physician for symptoms of RTIs and thus obtain help for the child. In Finland, parents also have the possibility to stay home from work with a sick child for up to 4 days. Another reason could be the attention the sick child receives from the parent, involving more verbal and child-directed stimuli. More child-directed speech, both in the amount and variation of language used, is associated with a larger vocabulary size in children [42,51]. This could particularly be the case for boys who, at an earlier stage in vocabulary growth, may benefit more from face-to-face talk compared to girls. One could also suggest that high levels of education and occupation of the parents (over 50% of the parents of boys with recurrent RTIs had high educational and occupational status) explain the vocabulary growth in boys with recurrent RTIs, as parents from higher SES backgrounds have been demonstrated to use strategies that support early language development. They use an abundant vocabulary with their children, ask more questions, and are less directory in their communication [52]. However, in the current study, boys with fewer RTIs at age 13–23 months did not have parents with significantly lower educational or occupational status. Another possible reason for the result could be the wide variation in children’s vocabulary during this time. The larger vocabulary growth could also be a result of confounding factors that we did not address in the current study.

In the present study, recurrent AOM was not significantly associated with smaller vocabulary growth. Our results align with previous studies finding minor, if any, negative associations between OM and language development [17,18,53,54,55]. However, the results contradict some earlier studies in which OM was associated with a smaller vocabulary size during the first two years [15,16]. Feldman et al. [15] found a weak significant correlation at two years of age between language level and the number of sick days due to OM with effusion during the second year or first two years of life. Haapala et al. [16] found restricted consonant inventories and a smaller vocabulary size at two years of age in children with recurrent AOM. In the study, hearing levels were assumed to be within the normal range, but fluctuating hearing loss was discussed as a possible reason for restricted language development [16]. Our present study was population-based, which could explain some of the differences from the findings of, e.g., Haapala et al. [16], which recruited participants upon presentation at a hospital for tympanostomy tube insertion because of chronic or recurrent otitis media.

### 4.3. Relation of Risk and Background Factors and Vocabulary Growth in Boys versus Girls

Significant differences in the occurrence of RTIs or AOM episodes in boys and girls, as suggested by Simoes [13] and Wang et al. [22], were not found in our study. In the current study, we found that vocabulary growth differed in relation to recurrent RTIs, high parental education and occupational levels, and vocabulary size at 13 months of age due to the child’s sex. None of the risk and background factors were associated with vocabulary growth between the ages of 13 and 24 months in girls. Differences due to the child’s sex in the trajectory of early vocabulary growth related to environmental factors have been little studied. There are studies showing that boys are at higher risk of RTIs [13,22] but no studies of RTIs in relation to vocabulary growth and the child’s sex. Similarly, the importance of parents’ educational and occupational status on the child’s language development has been established previously [29,31,41,56] but has rarely been discussed from a gender perspective.

In the present study, a high educational level of the father and high occupational status of the parents were significantly related to larger vocabulary growth but only in boys. Similar findings were found in a study by Barbu et al. [26], where low parental SES had a greater negative effect on language development in boys than in girls. It also supports findings by Lankinen et al. [27], where the paternal occupational level was more strongly associated with the expressive vocabulary size at 24 months of age in boys than in girls. Our study supports the findings of Barbu [26] and Lankinen [27] that found that early language development seems to be differently sensitive to environmental factors as a function of the child’s sex. The vocabulary growth of the girls was not associated with any risk or background factors, while the boys’ vocabulary growth was more sensitive to environmental factors. One possible explanation could be that girls, who are often more advanced in early vocabulary skills, have a more established language foundation compared to boys at a similar age and thus are not influenced by environmental factors in the same way as boys.

Further studies would gain from examining vocabulary growth at the same developmental stage in boys vs. girls in relation to environmental factors. In their study analyzing the development of lexical categories, Nylund et al. [28] found that different environmental factors influenced different lexical categories in boys vs. girls. The reason could be differences in the sensitivity of boys and girls to various environmental factors or that these were measured at the same age in boys and girls and not in the same developmental phase. Further studies are needed to determine possible underlying factors behind the differences in early language development between boys and girls. Moreover, Galsworthy et al. [57] concluded in their study that environmental and genetic factors can have explicit sex-dependent influences on early verbal development. The possible difference in how early vocabulary development in boys vs. girls is sensitive to environmental factors and what within these factors interacts with early language development needs to be addressed in further studies. Moreover, in assessing early language development, gender-specific norms could help to detect late talkers correctly, as combined norms can lead to the less accurate detection of, e.g., girls with limited language [58].

### 4.4. Strengths and Limitations of the Study

The strength of this study is its prospective birth cohort setting with the collection of meticulous infection data with daily notations of respiratory symptoms during the first two years of life and the prospective assessment of vocabulary growth from age 13 to 24 months. It also shows new findings on how early vocabulary development is associated differently with environmental factors depending on the child’s sex. However, there are some limitations that should be noted when interpreting the results. The mother’s educational level was higher in the included cohort than among those who were excluded. As a high socioeconomic background is associated with stronger language development in children, this should also be considered [20,31]. However, in the present study, the mother’s educational level was not associated with larger vocabulary growth in the children. Another confounding factor could be that parents who are interested in the development of their child chose to participate in the cohort study. The study used self-evaluation questionnaires to evaluate vocabulary development, which can be a risk for the reliability of the study. However, the CDI has been shown to be a reliable tool in early language assessment [59,60]. It is also possible that the CDI-I and CDI-T questionnaires were not completed by the same parent, which may have influenced how the vocabulary was measured. The present study was conducted in Finland, a European country with good health care and support systems for families. Therefore, for sociocultural and other reasons, the results may not be relevant to families living in other countries, for example, in the developing world. The diagnostic and treatment modalities of OM may be limited in developing countries, leading more frequently to complications, including hearing loss, with other possible effects on early language development than the results of the present study [61,62,63].

## 5. Conclusions

The present study shows that early vocabulary growth is not restricted by recurrent RTIs or AOM episodes. Interestingly, boys seemed to gain from episodes of RTIs, yielding a growth in vocabulary size, although not as a result of the infection itself. Instead, the results highlight that differences in how early language development interacts with environmental factors may be concealed if we use combined language measures for boys and girls. They thus corroborate earlier studies emphasizing sex differences. Moreover, our findings indicate that studying language development separately in boys and girls gives us a broader understanding of the factors influencing it. The clinical practice also benefits from more precise knowledge of the factors influencing early language development. In the current study, having siblings, staying in day care outside the home, and the mother’s level of education were not associated with vocabulary growth between 13 and 24 months of age, contradicting some earlier findings.

## Figures and Tables

**Figure 1 ijerph-19-15560-f001:**
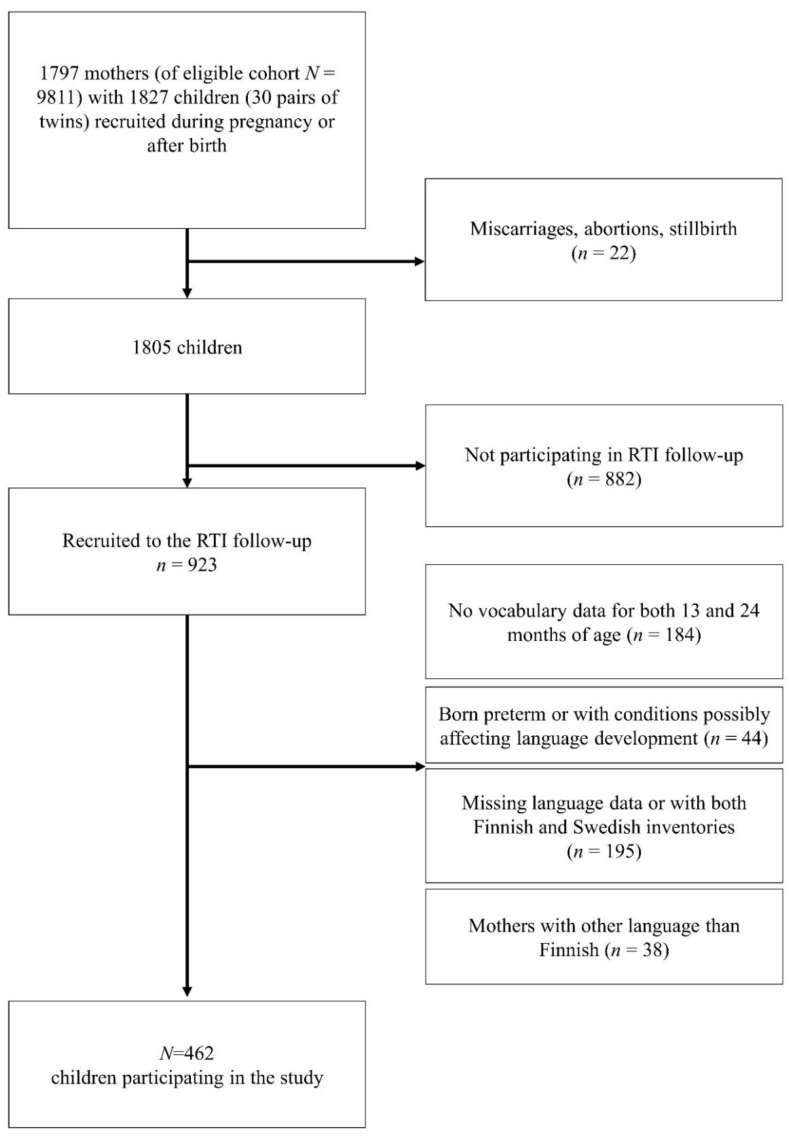
Flowchart of the recruitment procedure. RTI = respiratory tract infection.

**Figure 2 ijerph-19-15560-f002:**
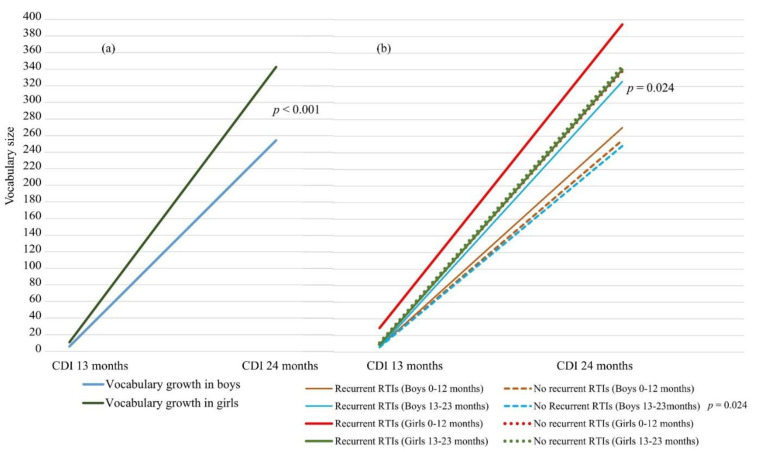
(**a**) Vocabulary growth in boys and girls between 13 and 24 months of age differed significantly (*p* < 0.001). (**b**) Mean vocabulary growth between 13 and 24 months of age in boys and girls with and without recurrent RTIs at 0–12 months of age and 13–23 months of age. CDI = MacArthur–Bates Communicative Development Inventory, RTI = respiratory tract infection. *t*-test for independent samples.

**Table 1 ijerph-19-15560-t001:** Descriptive background characteristics of the study population.

Characteristic	Included in the Study (*N* = 462)			
	Boys		Girls	
	*n*	(%)	*n*	(%)
Child’s sex (boys)	248	(53.7)	214	(46.3)
Firstborn				
yes	153	(61.7)	129	(60.3)
Age at birth, mean (SD)				
mother	31.2 (4.4)		31.1 (4.3)	
father	33.1 (5.4)		33.2 (5.5)	
Day care outside the home				
at 13 months	65	(26.2)	52	(24.3)
at 24 months	129/229	(56.3)	108/200	(54.0)
High educational level *				
mother	162/244	(66.4)	135/206	(65.5)
father	112/244	(45.9)	102/200	(51.0)
High occupational level **				
mother	135/214	63.1	113/180	(62.8)
father	106/197	(53.8)	103/170	(60.6)
Family income, average or more ***				
yes	101/245	(41.2)	105/209	(50.2)
Late onset of speech				
mother	3	(1.2)	3	(1.4)
father	4	(1.6)	3	(1.4)

* Bachelor’s, master’s, or doctoral degree. ** Professional. *** Income of EUR 3000 per month or more.

**Table 2 ijerph-19-15560-t002:** Comparison of vocabulary growth between 13 and 24 months of age in children with and without recurrent RTIs and AOM. *t*-test for independent samples.

Variable	Vocabulary Growth (Expressive Language)									
	Boys					Girls				
	*n*	*Mean*	*SD*	*p*	Cohen’s *d* [95% CI]	*n*	*Mean*	*SD*	*p*	Cohen’s *d* [95% CI]
Recurrent RTIs, 0–12 months				0.665	−0.09 [−0.49, 0.31]				0.312	−0.24 [−0.72, 0.23]
(≥91 days) yes	27	263.9	166.1			19	365.7	132.1		
no	210	249.3	163.6			188	329.5	150.2		
Recurrent RTIs, 13–23 months				0.024 ^a^	−0.48 [−0.90, −0.06]				0.970	0.01 [−0.45, 0.47]
(≥120 days) yes	25	320.3	159.9			20	331.6	147.0		
no	211	242.2	162.6			187	332.9	149.2		
Recurrent AOM, 0–12 months				0.089	−0.31 [−0.74, 0.12]				0.539	−0.16 [−0.69, 0.36]
(≥3 episodes) yes	23	289.2	125.6			15	359.1	145.4		
no	207	239.0	164.0			184	334.6	147.8		
Recurrent AOM, 13–23 months				0.477	−0.14 [−0.52, 0.24]				0.050	−0.34 [−0.82, 0.15]
(≥4 episodes) yes	30	268.5	167.8			18	378.7	92.5		
no	212	245.7	164.1			194	328.5	152.9		

Abbreviations: AOM = acute otitis media, RTI = respiratory tract infection. ^a^
*t*(234) = −2.275, *p* = 0.024.

**Table 3 ijerph-19-15560-t003:** Effects of risk and background factors on vocabulary growth separately in boys and girls between 13 and 24 months of age, as calculated with a linear regression analysis.

Variable	Expressive Vocabulary Growth in Boys and Girls									
	Boys					Girls				
	*B*	*SE B*	*β*	*p*	95% CI	*B*	*SE B*	*β*	*p*	95% CI
Expressive vocabulary size at 13 months	3.41	1.05	0.26	0.001	[1.34, 5.48]	2.23	0.84	0.25	0.009	[0.58, 3.88]
Presence of siblings	−15.93	27.56	−0.05	0.564	[−70.43, 38.56]	−59.18	27.70	−0.20	0.035	[−114.06, −4.30]
Recurrent RTIs, 0–12 months	−10.05	45.08	−0.02	0.824	[−99.19, 79.09]	33.84	50.30	0.07	0.503	[−65.83, 133.51]
Recurrent RTIs, 13–23 months	109.88	43.79	0.21	0.013	[23.29, 196.46]	−47.36	51.83	−0.09	0.363	[−150.07, 55.35]
Recurrent AOM, 0–12 months	10.34	43.38	0.02	0.812	[−75.40, 96.07]	13.60	54.76	0.02	0.804	[−94.91, 122.10]
Recurrent AOM, 13–23 months	34.26	36.35	0.08	0.348	[−37.61, 106,14]	68.84	45.74	0.14	0.135	[−21.81, 159.48]
High educational level of the mother *	−18.84	32.64	−0.06	0.565	[−83.38, 45.71]	−24.19	35.09	−0.08	0.492	[−93.73, 45.35]
High educational level of the father *	18.25	36.58	0.06	0.619	[−54.08, 90.59]	17.62	31.02	0.06	0.571	[−43.86, 79.10]
High occupational level of the mother **	44.47	33.40	0.14	0.185	[−21.58, 110.51]	5.68	32.75	0.01	0.863	[−59.22, 70.57]
High occupational level of the father **	35.81	35.61	0.11	0.316	[−34.60, 106.23]	−24.61	31.52	−0.08	0.437	[−87.08, 37.85]
Day care, 13 months	5.68	32.87	0.02	0.863	[−59.32, 70.68]	−8.60	33.48	−0.03	0.798	[−74.95, 57.75]
Day care, 24 months	28.42	26.84	0.09	0.292	[−24.66, 81.49]	45.81	28.66	0.16	0.113	[−10.98, 102.59]
	*F*(12, 138) = 2.70; *p* = 0.003; *R*^2^ = 0.19; *R*^2^*_Adj._* = 0.12					*F*(12, 111) = 1.52; *p* = 0.126; *R*^2^ = 0.14; *R*^2^*_Adj._* = 0.05				

* Bachelor’s, master’s, or doctoral degree. ** Professional. Abbreviations: RTI = respiratory tract infection, AOM = acute otitis media, *SE* = standard error.

## Data Availability

Not applicable.

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
