# Peer review of "Influence of Respiratory Tract Infections on Vocabulary Growth in Relation to Child’s Sex: The STEPS Study"

_ijerph, 2022, doi:10.3390/ijerph192315560_

Round 1

Reviewer 1 Report

Dear author(s),

I am honored to have the opportunity to review your manuscript. For a limited time, my review comments are my personal opinions and suggestions, and they are not always correct. Any of the comments are not intended to offend anyone.

Overall, this is an intriguing study that may be of interest to readers interested in language pedagogy and medical epidemiology. The author employs appropriate statistical methods, is fluent in English, and the manuscript is well-written. However, I have some personal concerns, the most pressing of which is whether there is a necessary link between RTIs and toddlers' early language development. As a consequence, even after reading it, pausing for a few days, and returning to it, I still have some reservations. I am aware that if RTIs are linked to early language development in toddlers, there will almost certainly be some debate or controversy. as we all know, children vary in their development of speech and language skills. However, they follow a natural progression or timetable for mastering the skills of language. After all, OM/AOM has received more attention in this field, and more research appears to link OM/AOM to "speech and language developmental milestones." "HEARING LOSS", you know.

I would like to give some personal advice here for the author(s)' consideration.

Major points:

1.     Given the obvious limitations of the study, I recommend that the authors limit the study to the country mentioned in the title. We make limited inferences based on limited evidence.

2.     I presume the authors should restructure the introduction, especially the first and second paragraph. The current evidence suggests that establishing a significant link between RTIs and vocabulary development is difficult.

Personally, 1) the second half of the first paragraph is somewhat confusing to the reader. It is suggested that one paragraph clarify a major point. 2) I believe that the effects of AOMs and RTI on vocabulary are not equivalent, but there may be an association. And the study's starting point is a lack of clarity about the relevant influences on early vocabulary growth.

Minor issues:

Line 43-44. The abbreviations RTIs and AOMs in this sentence may be confusing to the reader and should be rewritten.

Line 48-50. “we controlled for the presence of siblings and vocabulary size at 13 months of age.” This sentence does not appear to address the main point of the first paragraph. As readers, we would expect to see more emphasis on the significance of this subject in the first paragraph.

Line 75. “One possible explanation to the 75 inconsistent results can be methodological differences.” This statement is perplexing, and the evidence is weak.

Line 293. In the horizontal coordinates of Figures 2(a) and (b), the variable names are inconsistent.

Line 81. Gender differences are one of the main components and research hypotheses of this study. However, the phrases "no relevant studies" and "not separately for boys and girls" appear several times in the article. What should be clarified is the significance and necessity of studying gender differences (including the significance and value). Perhaps this is the most deserving basis for a study.

Line 91. Were the relevant components of SES examined in depth in this study, given that the authors stated that a low socioeconomic (SES) background influenced language development? Did the authors take into account the influence of socioeconomic status and other variables on the dependent variables?

Line 125-137. It is recommended that the authors express the "research hypothesis" more concisely.

Line 138-147. Given that this study is part of a larger earlier study, please clarify the relationship of this study to ref. [45]. May I ask the authors how representative the results are now that data from 2007 is being published?

Line 248. What I would like to know from the authors is whether there is a difference between SPSS 24.0 and 26. Is it necessary to process the data separately using two different versions of the same software? Perhaps reporting a single version is a better option?

With regards,

Reviewer

Author Response

Major points:

  1. Given the obvious limitations of the study, I recommend that the authors limit the study to the country mentioned in the title. We make limited inferences based on limited evidence.

Thank you for the suggestion! We found it some problematic to include the name of the country in the title of the manuscript but  included it in the abstract. In the limitation part of the study we also mention that the results from a study conducted in Finland may not be applicable in other countries due to socio-cultural differences.

  1. I presume the authors should restructure the introduction, especially the first and second paragraph. The current evidence suggests that establishing a significant link between RTIs and vocabulary development is difficult.

Personally, 1) the second half of the first paragraph is somewhat confusing to the reader. It is suggested that one paragraph clarify a major point. 2) I believe that the effects of AOMs and RTI on vocabulary are not equivalent, but there may be an association. And the study's starting point is a lack of clarity about the relevant influences on early vocabulary growth.

Thank you for this comment. Some of the text was mentioned without any previous introduction, which made it difficult to understand. We have now remodelled the introduction to be clearer.

Minor issues:

Line 43-44. The abbreviations RTIs and AOMs in this sentence may be confusing to the reader and should be rewritten. This sentence has now been removed

Line 48-50. “we controlled for the presence of siblings and vocabulary size at 13 months of age.” This sentence does not appear to address the main point of the first paragraph. As readers, we would expect to see more emphasis on the significance of this subject in the first paragraph. Thank you for pointing this out! The sentence was not appropriate and has now been removed. The factors mentioned in the sentence are presented later in the text.

Line 75. “One possible explanation to the 75 inconsistent results can be methodological differences.” This statement is perplexing, and the evidence is weak. Thank you, this was a vague statement as it was not properly referring to the discussion before it. It has been remodeled.

Line 293. In the horizontal coordinates of Figures 2(a) and (b), the variable names are inconsistent. Thank you, this has now been corrected

Line 81. Gender differences are one of the main components and research hypotheses of this study. However, the phrases "no relevant studies" and "not separately for boys and girls" appear several times in the article. What should be clarified is the significance and necessity of studying gender differences (including the significance and value). Perhaps this is the most deserving basis for a study. Thank you! This is absolutely right and has been emphasized more in the abstract and the introduction.

Line 91. Were the relevant components of SES examined in depth in this study, given that the authors stated that a low socioeconomic (SES) background influenced language development? Did the authors take into account the influence of socioeconomic status and other variables on the dependent variables? Thank you for the question. We are not sure that we understand it correctly, but our answers are here below. Before describing the articles of Lankinen and Barbu et al, we will just point out that these references are not mentioned here because of what SES parameters they have used, but because they have found differences in language development in relation to environmental factors as a function of the child´s sex, as it is mentioned there ”Nonetheless, a small number of studies have demonstrated differences in the influence of environmental factors on early language development as a function of the child´s sex.” The present study has in the same way as Barbu et al and Lankinen et al analysed language development separately in boys and girls (in relation to RTIs, day care, parental education and occupation etc.)

The authors Barbu et al. (2015) compared language development in families from high vs low SES, based on the family´s occupational status following the French National Institute of Statistics and Economic Studies. The criteria were: “high-SES parents belonged to group 3 (e.g., teachers and scientific professions, senior managers, engineers) and low-SES parents to group 6 (e.g., industrial, artisanal and agricultural workers and drivers). Both parents were from the same SES. When one of the parents was unemployed (i.e., did not work outside the household), only the occupation of the working parent was taken into consideration.

In the study of Lankinen et al (2018) the paternal education was considered high if he had completed a degree higher than a high school diploma. The occupational status was considered high if the father was working as an expert or in a leading position

In the present study we have analysed SES factors separately for the mother and the father but not as a combined measure as Braveman et al. (2005) and Duncan & Magnusson (2012) have suggested. The different levels for education and occupation were based on those used by the Statistics in Finland and the classification in high and low educational level and occupational status followed the same as in Lankinen et al. (2018).

Line 125-137. It is recommended that the authors express the "research hypothesis" more concisely. Thank you, the hypothesis has been rephrased.

Line 138-147. Given that this study is part of a larger earlier study, please clarify the relationship of this study to ref. [45]. May I ask the authors how representative the results are now that data from 2007 is being published? The present study is a sub study to the larger STEPS study with 1827 children (1797 families). From these families 923 children participated in an infection follow-up. From this group of children 462 children had Finnish vocabulary data for both 13 and 24 months of age. This is the sample of the present study.

The STEPS study is an ongoing longitudinal study, which was not mentioned in the methods but has been added. The language data is from the first and the second year of the child (i.e., years 2008–2012). This is a large data and still representative on group level.

Line 248. What I would like to know from the authors is whether there is a difference between SPSS 24.0 and 26. Is it necessary to process the data separately using two different versions of the same software? Perhaps reporting a single version is a better option? Thank you for the question. As we have been working on this data for some time, we have just mentioned the different versions that has been used as the program has been updated. But it is okey to just mention the last version.

With regards

Reviewer 2 Report

This is a very interesting paper that investigated the influence of respiratory tract intentions on the vocabulary growth of infants aged 13-24 months, and compared the differences between boys and girls. The methods are on the whole reliable. The findings are faithfully reported. All in all, this paper is almost qualified for publication. However, three problems need to be solved.

 1. Why do you consider the background factors? I know that these educational and social background factors may contribute to vocabulary growth, but you need to convince the readers that these factors are worth studying. In addition, the title does not cover the content of background factors. I suggest a moderation between the title and the content.

 2. The linear regression models are inappropriate, for some of the independent variables, such as the educational and occupational levels, are discrete ones. I suggest using dummy variables instead.

 3. I also suggest a round of revision of the English language used by a native speaker. I found some expressions not native-like or grammatically acceptable while reading, e.g. “According to the results children suffering from recurrent RTIs did not on a group level have fewer words at 13 or 24 months of age than children less sick.” (line 66), “For two of the study children” (line 180), “The parents were advised to visit the STEP-study clinic during RTIs” (line 187), “The following child and parental background data that has been linked to early language development was included in the study” (line 209), and “There is also a possibility that not the same parent completed the CDI-I and CDI-T questionnaires which may have influenced how vocabulary was measured.” (line 526).

Author Response

  1. Why do you consider the background factors? I know that these educational and social background factors may contribute to vocabulary growth, but you need to convince the readers that these factors are worth studying. In addition, the title does not cover the content of background factors. I suggest a moderation between the title and the content. Thank you for the comment. The background factors are included because there is evidence that they may influence language development, but they are not the main theme of the study which is the reason they are not mentioned in the title and not emphasized as much as the risk factors.
  2. The linear regression models are inappropriate, for some of the independent variables, such as the educational and occupational levels, are discrete ones. I suggest using dummy variables instead. Although some of our independent variables are discrete, they are, however, variables of ordinal scale, and that's why we decided to use them in regression analysis in their original form, because we were mainly interested in the overall effect of a background variable (education) instead of the pairwise differences of the effects.

  1. I also suggest a round of revision of the English language used by a native speaker. I found some expressions not native-like or grammatically acceptable while reading, e.g. “According to the results children suffering from recurrent RTIs did not on a group level have fewer words at 13 or 24 months of age than children less sick.” (line 66), “For two of the study children” (line 180), “The parents were advised to visit the STEP-study clinic during RTIs” (line 187), “The following child and parental background data that has been linked to early language development was included in the study” (line 209), and “There is also a possibility that not the same parent completed the CDI-I and CDI-T questionnaires which may have influenced how vocabulary was measured.” (line 526). Thank you for this remark. The text has been revised by a native English editor.

Round 2

Reviewer 1 Report

The manuscript was improved and all my concerns were addressed by the authors.